# A Comparative Study of the Resilience of Urban and Rural Areas under Climate Change

**DOI:** 10.3390/ijerph19158911

**Published:** 2022-07-22

**Authors:** Qingmu Su, Hsueh-Sheng Chang, Shin-En Pai

**Affiliations:** 1School of Architecture and Planning, Fujian University of Technology, Fuzhou 350118, China; martain@foxmail.com; 2Department of Urban Planning, National Cheng Kung University, Tainan 70101, Taiwan, China; p26074193@mail.ncku.edu

**Keywords:** resilience indicator, climate change, evaluation system, urban–rural differences, binary logistic regression

## Abstract

The impact of climate change in recent years has caused considerable risks to both urban and rural systems. How to mitigate the damage caused by extreme weather events has attracted much attention from countries in recent years. However, most of the previous studies on resilience focused on either urban areas or rural areas, and failed to clearly identify the difference between urban and rural resilience. In fact, the exploration of the difference between the resilience characteristics of cities and villages under climate change can help to improve the planning strategy and the allocation of resources. In this study, the indicators of resilience were firstly built through a literature review, and then a Principal Component Analysis was conducted to construct an evaluation system involving indicators such as “greenland resilience”, “community age structure resilience”, “traditional knowledge resilience”, “infrastructure resilience” and “residents economic independence resilience”. Then the analysis of Local Indicators of Spatial Association showed some resilience abilities are concentrated in either urban or rural. Binary logistic regression was performed, and the results showed urban areas have more prominent abilities in infrastructure resilience (the coefficient value is 1.339), community age structure resilience (0.694), and greenland resilience (0.3), while rural areas are more prominent in terms of the residents economic independence resilience (−0.398) and traditional knowledge resilience (−0.422). It can be seen that urban areas rely more on the resilience of the socio-economic structure, while rural areas are more dependent on their own knowledge and economic independence. This result can be used as a reference for developing strategies to improve urban and rural resilience.

## 1. Introduction

Global warming in recent years has triggered extreme weather events across the world. Particularly, the intensity and frequency of typhoons and rainfall are increasing, which intensifies the risk of disasters to life and property. How to reduce the risks brought by climate change has been one of the important issues that have attracted much attention in spatial planning and disaster management [1,2]. In order to mitigate the risk of extreme weather events caused by climate change, the United Nations Office for Disaster Risk Reduction (UNDRR) has proposed resilience as an important concept for the reduction in the harm of disasters. So far, the concept of resilient cities and resilience planning has received wide attention from international government organizations and the urban planning field. It has gradually transformed from an emerging research issue to a dominant research direction [3,4]. The so-called resilience refers to the ability to cope with the threat of external events under the existing internal conditions of the human system. The field of ecology first introduced the concept of resilience, defining it as “a measure of the persistence of a system and its ability to absorb change and disturbance” [5]. Since then, many concepts of resilience have been derived. The concept of resilience is sometimes used as a measure of the ability of a system to recover after an event occurs, and sometimes it is regarded as the speed at which the system returns to its original state when an event occurs. However, it is generally the ability of internal systems to resist, buffer disturbances, absorb disturbances, self-organize, learn and adapt in response to external shock events [6,7]. Therefore, this study focuses the concept of resilience on the ability of human systems to withstand the threat of external events under existing internal conditions.

However, the previous discussion on the resilience strategy to cope with climate change mainly focuses on urban spaces or densely populated areas [8], while ignoring rural areas that are also threatened by climate change. Therefore, there is still a lack of understanding of the risks and corresponding strategies in rural areas. In fact, cities and rural areas have significantly different resilience in society, economy, environment, and infrastructure when facing disaster risks caused by climate change [9]. However, most of the previous studies on urban resilience explored the ability of the economy and industry of urban systems to respond to the threat posed by climate change, while most of the studies on rural resilience focused on the ability of the rural areas to survive under climate change. The external events will bring the same impact to both the urban and rural areas. Therefore, building an indicator framework of resilience that adapts to the cities and villages will help to identify the difference in resilience between the urban area and rural area, and will facilitate the future allocation of related resources and improve strategies. The research on urban and rural resilience is also a strategic exploration to cope with the different responses of urban and rural areas to disasters and corresponding mechanisms under climate change.

Cities have functions to meet the various needs of local residents, and climate change will cause an inevitable impact on the functions that the urban system provides [10]. These functions include social and economic functions, infrastructure, environment, and anti-disaster facilities. This is also the main basis for scholars to establish the indicators of resilience. However, for rural areas, in addition to their basic functions, the experience left over from history also plays an important role in coping with climate disasters [11]. Therefore, traditional knowledge is also a major factor in dealing with disasters.

In terms of resilience assessment, there is still a lack of consensus on the measurement methods and operations of resilience. The Disaster Resilience of Place Model proposed by Cutter et al. (2008) is mainly to quantify the spatial resilience of selected places, which can be classified as a local resilience model; the Coupled Social-Ecological Metrics model mainly introduces the tools of system dynamics and complexity to analyze the resilience of the community; the Teleconnection Metrics model mainly hopes to solve the problem of nonlinear dynamics in the Nested System, so it analyzes through different geographical locations to explore how the community is related to factors in different geographical locations through long-distance connections, but this method is mainly qualitative evaluation [12,13]. Taken together, community resilience is deeply affected by vulnerability, global climate change, and natural disasters, resulting in each knowledge area having its own research framework and a set of ways to conceptualize community resilience. As a result, there is currently a lack of consensus on quantitative methods for assessing community resilience. Since this research hopes to establish resilience indicators that can be used to evaluate local urban and rural areas, the construction of the resilience indicators will be based on the local resilience indicators, and the advantages of various evaluation methods will be integrated to establish an evaluation model of urban and rural resilience.

The huge difference between urban and rural spatial patterns makes it particularly crucial to explore the difference between urban and rural resilience in that spatial unit. Although relevant studies have conducted a comprehensive evaluation of resilience at a single spatial level [6,14], the spatial scale is too high to clearly identify the resilience of local communities in the face of climate change, and even the urban and rural areas cannot be clearly distinguished in the spatial unit. Therefore, it is impossible to obtain the difference between urban and rural resilience. Through literature review, this study explored the risks that cities and villages face under climate change and the key factors affecting their resilience and then constructed the indicators of resilience that can evaluate cities and villages under climate change. After that, spatial autocorrelation was used to identify whether the resilience in the research area is spatially related, and to identify whether the hotspots are concentrated in urban or rural areas. Finally, according to the results of spatial autocorrelation, binary logistic regression was conducted to find the differences between urban and rural resilience. The analysis results can be used as guidance for the future strategy of improving urban and rural resilience.

## 2. Methodology

### 2.1. Research Framework

In this study, a three-stage research framework was constructed (Figure 1). In the first stage, an indicator system of resilience was built based on urban and rural areas. At this stage, the indicators of urban and rural resilience in the past studies were comprehensively analyzed through a literature review, and the corresponding dimensions of the indicators were determined. However, since the indicators of resilience in this study were established through a literature review, the impact of the indicators on resilience is highly correlated. Therefore, Principal Component Analysis was used to reorganize the principal component of the originally selected indicators, analyze the relationship behind the indicators of resilience, and then establish the indicator structure of resilience.

The second stage is the spatial autocorrelation analysis of the indicators of resilience. First, whether different resilience abilities are spatially correlated was explored through Moran’s I. Then, the spatial distribution patterns of different resilience abilities were identified through LISA, so as to investigate whether the spatial distribution of different resilience abilities is the same, and whether some spatial characteristics are concentrated in urban or rural areas.

In the third stage, urban and rural resilience was compared. The characteristics of urban and rural space units on different indicators of resilience were compared according to the results of the second stage through binary logistic regression, so as to explain and discuss the analysis results. In the operation of the binary logistic regression model, through the establishment of a dummy variable, this study first defined the smallest statistical area belonging to a city in the spatial units as 1, and defined the smallest statistical area belonging to the rural area as 0. Second, the iterative history was analyzed by SPSS software, and Maximum Likelihood Estimation was used to obtain the optimal parameter values. Finally, the accuracy of the model was verified through the Omnibus model coefficient, Nagelkerke *R*^2^ and Hosmer–Lemeshow. According to the regression coefficient, the relationship between each independent variable and the dependent variable can be confirmed, and then the category of the indicators belonging to the urban area or rural area can be acquired so that the difference in resilience between the city and village can be identified.

### 2.2. The Research Area and the Identification of Urban and Rural Spatial Units

Due to the complex terrain and special geographical location, Taiwan is severely threatened by the extreme weather events caused by climate change, such as flood disasters in coastal areas and the debris flow in mountainous areas [15]. In this case, to study local resilience, the research scope must cover various risk areas that may face extreme climate events with diverse terrains. Chiayi County in Taiwan is an area that is severely affected by natural disasters, thus it is suitable for the research. At the same time, in order to compare the characteristics of the urban and rural areas on each resilience indicator, the statistical unit of urban and rural spaces should be defined first. In terms of the selection of cities, the urban planning district of Chiayi County was used as the standard, which defines the scope of the metropolitan area. Therefore, we select its community boundary as the statistical unit. Regarding the selection of villages, the rural area in the non-urban land use zoning was used as the standard. Its original definition standard is based on the current situation of use. Therefore, the minimum statistical area is mainly based on the community boundaries and village boundaries defined by the government. Finally, a total of 512 smallest statistical areas were selected as rural areas, and 1133 smallest statistical areas were selected as cities. The results can be used as the statistical units in the subsequent binary logistic regression model to compare the difference in the indicators of resilience between the urban and rural areas after Principal Component Analysis (Figure 2).

### 2.3. A Comparative Approach to Urban–Rural Resilience Differences

#### 2.3.1. Principal Component Analysis (PCA)

Principal Component Analysis was used to process the content of the indicators to make the indicators of resilience more complete. Principal Component Analysis is mainly used to analyze the correlation between variables, and then reduce the original number of variables to generate new variables [16,17]. This process is called dimension reduction, and the new variable obtained is the principal component. If the original variables are not correlated, Principal Component Analysis cannot reduce the number of variables at all. Only when the variables are highly correlated with each other can the number of variables be simplified, and the stronger the correlation, the higher the simplification of the variables [18]. We used Statistical Product and Service Solutions (SPSS) software for analysis.

#### 2.3.2. Local Indicators of Spatial Association (LISA)

The analysis of Local Indicators of Spatial Association is a basic concept of Global Moran’s I that extends spatial autocorrelation. It mainly compares whether adjacent attributes are close to each other by directly calculating the difference in attribute values [19]. This study uses LISA to identify the spatial distribution patterns of different resilience abilities, and then explores the spatial differences of different resilience abilities. Based on the results, binary logistic regression can be performed on the urban and rural spatial units to examine the difference between urban and rural resilience. The calculation of the correlation between the space units of LISA and the surrounding space is shown in Formula (1):(1)$$I_{i}=Z_{i}\underset{j}{\sum }W_{\mathrm{ij}}Z_{j}$$
where $I_{i}$ is the value of Local Moran’s I; z represents the spatial attribute relationship between two adjacent regions; $Z_{i}$ is the value of z of $X_{i}$; $Z_{j}$ is the value of z of $X_{j}$; $W_{\mathrm{ij}}$ is the spatial weight matrix between the research objects i and j.

When the significance reaches the standard and Local Moran’s I is positive, if $Z_{i}>0$, it means that the observed value of location i and the adjacent areas is relatively high, which is expressed as a high-high district; if $Z_{i}<0$, it indicates that the observed value of location i and the adjacent areas is relatively small, which is represented by the low-low district in this study. These two results indicate that a certain area is positively correlated with its neighboring areas. On the contrary, when the significance reaches the standard and Local Moran’s I is negative, if $Z_{i}>0$, it means that the observed value of position i is much higher than that of the adjacent areas, which is expressed as a high-low district; if $Z_{i}<0$, it means the observed value of position i is much lower than that of the adjacent areas, which is a low-high aggregation and is expressed as a low-high district in this study. These two results indicate a negative spatial correlation [20].

#### 2.3.3. Binary Logistic Regression

This study aims to compare urban and rural resilience. Therefore, if the urban and rural areas are defined as binary in the dependent variable, with the help of LISA’s spatial analysis results and binary logistic regression analysis, it is possible to distinguish the influence of the independent variables of each resilience characteristic on the urban and rural areas, and then produce the analysis results that can facilitate the comparison of the characteristics of urban and rural resilience. Logistic regression adopts the Maximum Likelihood Estimation (MLE) to maximize the probability of observation of the variable in the result to obtain the optimal estimate of the independent variables [21,22]. Logistic regression is a probability model in which the dependent variable (p) varies between 0 and 1. When multiple independent variables are added, the Formula (2) of logistic regression can be expressed as follows:(2)$$\mathrm{Logit}\;p=\ln\frac{p}{\left(1-p\right)}=\sum b_{i}x_{i}$$
where b_i_ is the influence coefficient, and x_i_ is the indicator of resilience.

Compared with the previous studies that only conducted comprehensive score evaluation or the comparison of the overall difference between urban or rural resilience, this study used the binary logistic regression model to further explore the aspects where urban and rural resilience is strong or weak.

## 3. Results Analysis

### 3.1. The Construction of the Indicator System and Principal Component Analysis

With reference to the existing literature and the framework of CIMO (context–intervention–mechanism–outcome), and based on the principles of scientificity, objectivity, comprehensiveness, and data availability, this study initially established an urban and rural disaster resilience indicator system [23]. In order to eliminate the subjectivity of index selection decisions, we refer to three databases, CNKI, Web of Science, and ScienceDirect, and select index elements with high frequency in recent years. These indicators are further selected by consultants and government staff in the fields of resilient cities, disaster risk, and urban–rural development, and are constructed in two dimensions: urban resilience and rural resilience (specific indicators are shown in Appendix A).

The key factors affecting local resilience under climate change were analyzed through a literature review. Since this research focuses more on urban and rural resilience to cope with climate change, rather than resilience in the face of disaster risk, the indicators of resilience in the category of non-climate change were excluded. Regarding the indicators of urban resilience, most studies believe that the indicators of resilience should involve the abilities of the city to alleviate the impact caused by disasters. Two dimensions, society and economy, are regarded as one of the indicators in all studies. These two indicators play a key role in urban resilience [8,24]. Some studies presented a view that the existence of resilience cannot be separated from the process of social operation, thus the natural environmental system, social environmental system, and built environmental system are all interconnected [13,25]. For example, Yoon et al. (2016) established six dimensions, including people, society, economy, environment, disaster prevention system, and urban space, and then assessed the resilience of communities in the face of disasters [6]. The key influencing factors of urban resilience include infrastructure and environmental factors [6,26]. In addition, since this study established the indicators of resilience under climate change, disaster threat is also a key factor [14,27]. For example, Maziar Yazdani et al. studied the impact of flood risk on medical infrastructure and proposed a new modeling framework to improve the resilience of medical infrastructure to floods [28,29]. In terms of the indicators of rural resilience, in addition to the indicators of urban resilience, two other aspects are particularly emphasized. First, the socio-economic background plays the most important part. To improve the resilience or adaptability of mountainous villages in the context of climate change, the primary method is to stimulate the economic development of the villages, so as to enhance the economic income of the residents. Especially, attention should be paid to how to reduce the dependence on agriculture and create more diversified income sources for mountainous villages [24,30]. Therefore, “the number of agricultural households” and “residents’ income” were incorporated into the economic indicators. Second, most rural areas make production decisions based on years of empirical observations, which can be called a decision-making process based on a traditional knowledge system [11]. Rural residents often use the life experience and knowledge inherited from their ancestors to deal with the impact of external climate change events [11,31]. For example, the study by Altieri et al. (2017) found that traditional knowledge can effectively reduce the impact of climate change on local residents and maintain a certain amount of agricultural products. This also means that traditional knowledge for villages will help improve their resilience [11]. Therefore, traditional knowledge is also a key factor affecting resilience. Most research results showed that the traditional knowledge possessed by the indigenous population can provide strategies to cope with climate change or disasters [31,32]. Hence, relevant indicators of the indigenous people were included in the indicators of resilience. In summary, this study established an initial indicator system of resilience, which contains 22 indicators in six orientations, including society, economy, infrastructure, environment, disaster threat, and traditional knowledge (Table 1).

In this study, 22 indicators were input into SPSS software for analysis (see Appendix A for specific calculation methods and data sources of indicators). First, Kaiser–Meyer–Olkin (KMO) (0.767) and Bartlett’s Sphericity Test were performed on the original indicators, and the results revealed there are common factors between the original indicators, thus factor analysis can be conducted. Then Principal Component Analysis was performed, and the principal components were extracted with the eigenvalue of 1 as the standard. Finally, five principal components were extracted, with the explanatory variable of 20.313%, 9.853%, 8.330%, 6.920%, and 5.002%, respectively. The total explanatory variable is 50.418%.

(1)Principal component 1: Greenland resilience

The analysis result is highly positively correlated with such original indicators as “agricultural land area”, “green area”, “population density” and “impervious area”, and is highly negatively correlated with “school” and “road density”. The low population density of the area leads to the lack of infrastructure. Therefore, the greenland and agricultural land of the local community have not been developed and became the resilience ability on which the area heavily relies. Thus, this principal component was named greenland resilience.

(2)Principal component 2: Community age structure resilience

The analysis result shows a high positive correlation with the original indicator, “dependency ratio”, and exhibits a high negative correlation with “education level” and “household size”. This resilience ability shows that due to the small dependency ratio of the area, the household size is bound to be smaller than that of the three-generation community. When the community with such an age structure faces the risks caused by climate change, the loss caused by external events can be reduced owing to a small number of senior citizens, thereby increasing community resilience. Thus, this principal component was named “community age structure resilience”.

(3)Principal component 3: Traditional knowledge resilience

The analysis result is highly positively correlated with the original indicators such as “percentage of indigenous population” and “proportion of aboriginal elderly population”. Principal component 3 mainly shows that the indigenous population who have lived in the area for a long time can use the knowledge inherited from their ancestors or their own accumulated experience to deal with natural disasters, and formulate corresponding strategies. Therefore, this principal component was named “traditional knowledge resilience”.

(4)Principal component 4: Infrastructure resilience

The analysis result is highly positively correlated with the original indicators: “residents’ income”, “medical facilities”, “fire station”, and “stratum subsidence”. This principal component indicates that since the local residents have a high economic level, they possess more choices regarding residences, and the economic level can also provide funding for the construction and maintenance of local community infrastructure. It can be seen that this principal component mainly represents the degree to which the infrastructure is complete, hence it was named “infrastructure resilience”.

(5)Principal component 5: Residents economic independence resilience

The analysis result is highly positively correlated with the original indicator “low-income households”, and is highly negatively correlated with “green infrastructure”. This principal component indicates that the lack of local planning and establishment of green infrastructure has resulted in the absence of the ability to deal with the risks of climate change. Therefore, local residents must rely on their own economic independence. This principal component was named “residents economic independence resilience”.

### 3.2. Spatial Autocorrelation Analysis of the Indicators of Resilience

The spatial autocorrelation analysis of the indicators of resilience demonstrated that Moran’s I exceeds 0.26, and aggregated distribution is presented. In order to understand the spatial distribution mode of each resilience indicator, this study used the univariate spatial autocorrelation analysis in the GeoDa software to identify the spatial correlation between the indicators of resilience. The local spatial autocorrelation results of the five indicators of resilience constructed in this research are all different. The detailed results are shown in Figure 3.

In terms of the spatial distribution pattern of greenland resilience (Figure 3a), most of the non-mountainous and coastal areas have high resilience scores of spatial units and adjacent spatial units. In addition, among the space units in the coastal area, many space units and their adjacent space units have low values. In the resilience score of community age structure resilience (Figure 3b), most of the regions with high values of spatial units and adjacent spaces are located in the west, and some spatial units in the eastern mountainous areas also exhibit such characteristics. In the resilience score of traditional knowledge capacity (Figure 3c), the spatial units in the eastern mountainous area and the adjacent spatial units show a distribution pattern of high-value concentration. In addition, the spatial distribution patterns with low toughness scores and adjacent spatial units with low values are mostly located in the western coastal areas. In the score of infrastructure resilience (Figure 3d), the areas with high values of both spatial units and adjacent spatial units are mostly concentrated in the urban areas near the west. The spatial units with lower resilience scores and the adjacent spatial units with low values are located in the western coastal and eastern hilly areas. Among the resilience scores of residents economic independence ability (Figure 3e), those with high resilience scores and their adjacent spatial units are mostly concentrated in the eastern non-mountainous areas. Those with low resilience scores and their adjacent spatial units are mostly located in the western coastal areas. The analysis results of spatial correlation showed that there may be differences in the resilience abilities between cities and villages. Therefore, it is necessary to further explore the differences between urban and rural resilience.

### 3.3. Analysis of the Difference between Urban and Rural Resilience

The spatial autocorrelation analysis revealed that the five types of resilience abilities are all spatially concentrated in high values. In this study, the ArcGIS software was used to select the urban and rural statistical units, and then the binary logistic regression in the SPSS software was adopted to explore the difference between urban and rural resilience in different dimensions. The results of the analysis are shown in Table 2. The accuracy of the binary logistic regression model used in this study is 41.1% (Nagelkerke *R*^2^ = 0.411), indicating that the predictive analysis accuracy of this model was the best at 41.1%. The Hosmer–Lemeshow test is mainly used to test the fit of the model. The significance of the test result in this study is 0.408 > 0.05, which means that the hypothesis of good fit of the regression model is accepted. The categories of the statistical units of principal component 1, principal component 2, and principal component 4 are prominent in urban areas, while the statistical unit categories of principal component 3 and principal component 5 are more obvious in rural areas.

(1)Infrastructure resilience

Infrastructure resilience shows positive loading, and residents’ income, stratum subsidence, fire station, and medical facilities have a significant contribution. Part of the funds for improving community infrastructure come from the taxation of local residents, and part of the funds for the construction or maintenance of medical facilities/fire stations even come from donations from local residents. Under this condition, the maintenance of infrastructure is highly dependent on the economic ability of local residents. The higher the residents’ income, the more helpful it is to the construction and subsequent maintenance of infrastructure. In areas where the stratum subsidence is severe, the land with a relatively stable geological space is mainly selected for urban development, and there are few areas for agricultural use in the subsequent industrial strategy of urban planning. To sum up, since cities are not threatened by severe stratum subsidence and have infrastructure supported by local residents’ income, the resilience of urban areas in the face of climate change has been strengthened.

(2)Community age structure resilience

In terms of the community age structure resilience, the indicators with large loading include education level, household size, and dependency ratio. The loading of education level and household size is negative, and the loading of dependency ratio is positive. The analysis results are relatively consistent with the age structure of communities and households in urban areas. In fact, since the 1960s in Taiwan, with the development of urbanization, the household structure has gradually changed from a large family where dozens of people lived together to a small family with only about four people, that is, the size of urban households is relatively small. As the size of the urban households shrinks, the proportion of elders in most households gradually decreases, and the dependency ratio of the community is reduced correspondingly. Generally speaking, the family structure and community age structure in urban areas have a more significant impact on the degree of resilience than those in rural areas.

(3)Greenland resilience

Regarding greenland resilience, the indicators with obvious loading include population density, agricultural land area, school, and road density. The resilience ability of this principal component is relatively inadequate due to the insufficient community infrastructure such as school and road density. Therefore, the resilience ability is more dependent on the small population density and green space. The analysis results showed that the greenland resilience in urban areas is more significant than that in rural areas. In urban areas, the greenland that was previously regarded as less economically valuable has increased the resilience of cities under the threat of climate change. However, because there is less greenland in rural areas, the greenland does not help to improve the resilience of rural communities under the threat of climate change.

(4)Residents economic independence resilience

In terms of the residents economic independence resilience, the indicators with prominent loading in this principal component are low-income households and green infrastructure. This ability is mainly used to explore how local residents can strengthen their resilience through their own economic independence in the absence of green infrastructure. The analysis results showed this type of resilience ability is more prominent in rural spaces. It can be seen that although rural areas are close to farmland or forests, there is no green infrastructure in rural space units, leading to a lack of the green base to provide environmental resilience. In this case, it can be said that rural residents are quite dependent on their own economic level due to the absence of planning for green infrastructure.

(5)Traditional knowledge resilience

Regarding the traditional knowledge resilience, the indicators with significant loading in this principal component include the percentage of the indigenous population and the proportion of the aboriginal elderly population. Since the indigenous population has lived in this area for a long time, they have more prominent sensitivity to local climate change, resource utilization, and familiarity with geographic space than ordinary people. At the same time, the aboriginal elders can help to disseminate and inherit traditional knowledge. The analysis results showed that rural areas possess prominent resilience ability in terms of traditional knowledge, which can also be found from the spatial distribution pattern in the spatial autocorrelation analysis. The research scope mainly covers the places where a large indigenous population gathers, and most of these places are rural areas. Therefore, rural area has more significant traditional knowledge resilience than urban area.

## 4. Discussion

Most of the previous studies on resilience emphasized the influence of infrastructure and socio-economic factors, thus these factors were often selected for exploring resilience [6,33]. However, after a literature review, it was found that traditional knowledge plays a significant role in coping with climate change in certain areas. Our research results also suggested that rural areas have more prominent resilience than urban areas in this respect. It can be seen that due to the lack of strong social and economic background, or the lack of complete infrastructure, the resilience of the rural areas heavily depends on the traditional knowledge of local residents.

In previous research, Tobin (1999) took Florida as the research object, which mainly measured the degree of internal resilience of the state after the impact of Hurricane Andrew [34]. However, it does not have a corresponding practice site but is based on qualitative analysis. This study combines local characteristics and provides the possibility to assess urban and rural resilience. Second, Mayunga’s (2007) study further extended to various disaster types and established a Community Disaster Resilience Index [35]. However, in resilience assessment, it is mainly discussed from the perspective of economic capital, but it ignores resilience capabilities other than environmental or social capital. A major feature of this study is that it expands the concept of resilience and establishes an evaluation system for urban and rural resilience from social, economic, environmental, and traditional knowledge. Then Yoon et al. (2016) also established six dimensions including people, society, economy, environment, disaster prevention system, and urban space based on the Community Disaster Resilience Index, and conducted resilience measures for 229 basic self-governing groups in South Korea [6]. However, the blind spot lies in the failure to clearly identify the resilience of local communities for discussion, while this study is based on local resilience construction and has wide practical implementation. Research by Hudec, Reggiani, and Šiserová (2018) developed the Resilience Capacity Index to measure the resilience or rebound of regions from climate change shocks, but it also failed to clearly identify the resilience of local communities [8]. To sum up, there is a lack of application of urban and rural resilience indicators in previous resilience index assessments. The establishment of indicators for key factors affecting urban and rural resilience in this study is an important step to improve localized urban and rural resilience.

Regarding the formulation of strategies for cities to respond to climate change, most of the previous research focused on reducing the impact of climate change through infrastructure planning [36,37]. The analysis results of this study also confirmed the importance of infrastructure to urban resilience, but on the other hand, it was also found that green space is also one of the key factors of urban resilience. In fact, green space in the city used to be regarded as land without economic value, but it has become a key factor affecting urban resilience under the threat of climate change. Therefore, how to increase the proportion of urban green space can be used as one of the important strategies for each city and county to establish spatial planning. Although it is also emphasized that infrastructure is an essential strategy for improving rural resilience, the long-term lack of a comprehensive planning strategy in rural areas makes it difficult to increase the construction of infrastructure. According to the analysis results of this study, the resilience abilities of the rural areas mainly depend on the economic independence of local residents and traditional knowledge at a non-hardware level. Therefore, in spatial planning, how to effectively use these abilities is the key to the subsequent overall planning of rural resilience, and it is also the foundation of maintaining rural resilience before the infrastructure is improved in rural areas. To sum up, when planning the location of infrastructure in the future, it will help to reduce the damage caused by the city if we can consider the characteristics of infrastructure for the city in coping with climate change and pay attention to its co-benefits. Villages have long lacked sound planning strategies and should rely more on their own resilience (such as traditional knowledge) to cope with climate change.

The biggest contribution of this research is the construction of an indicator framework of resilience that adapts to urban and rural areas. This framework helps to explore the differences between the corresponding resilience characteristics of cities and villages under the threat of climate change. By integrating the indicators of urban and rural resilience, the indicator system is more applicable. Through the Local Indicators of Spatial Association, this framework established the correlation between the indicators of resilience regarding spatial distribution, so as to understand the aggregation in the urban or rural areas. In addition, a binary logistic regression model was used to explore the different resilience capabilities of urban and rural areas more thoroughly.

There are still some limitations in this research. (I) Since the data were published in different years, the analysis based on the latest related data cannot facilitate the comparison of the changes in urban and rural resilience in terms of time under the influence of climate change. If more time-section data can be used, it will help to obtain more accurate resilience capabilities, and thus deepen the understanding of the differences in resilience between urban and rural areas. (II) The spatial units compared in this study are limited to urban and rural areas formulated in the research design, and many areas that are not located in the above spatial units are ignored. Therefore, it is impossible to compare various types of national land spaces. (III) The results of this study are obtained based on correlations and have certain limitations. Follow-up research can also help analyze the presentation of results from the “causal inference” approach.

## 5. Conclusions

Climate change has imposed a great impact on both urban and non-urban systems, but under the threat of climate change, the resilience characteristics of urban and rural areas are quite different [38,39]. Through the establishment of the indicator framework of resilience, the following conclusions were obtained:

(I) Based on the literature review, this study selected the indicators of resilience suitable for evaluating both urban and rural areas and used Principal Component Analysis to reduce dimensions. Finally, five principal components were extracted, namely, greenland resilience, community age structure resilience, traditional knowledge resilience, infrastructure resilience, and residents economic independence resilience. (II) In this research, spatial autocorrelation was conducted to analyze and explore the spatial autocorrelation of resilience abilities. The spatial location of the hot and cold spots of the various resilience abilities derived from the analysis can provide reference for relevant government departments. (III) A comprehensive analysis of the five types of resilience abilities found that under the threat of climate change, cities are relatively dependent on the infrastructure built with local resource input and the resilience provided by the environment. The rural areas are relatively dependent on the economic abilities of the local residents themselves or the resilience at the level of knowledge application. This is also the biggest difference between the urban and rural areas under the threat of climate change.

The comparison of urban and rural resilience abilities can help to develop stronger targeted measures in the four stages of disaster management: disaster reduction, disaster preparedness, response, and recovery. However, it needs to be emphasized that the research on resilience is still in a preliminary stage regarding the handling of climate change and risk management, and further efforts need to be invested.

## Figures and Tables

**Figure 1 ijerph-19-08911-f001:**
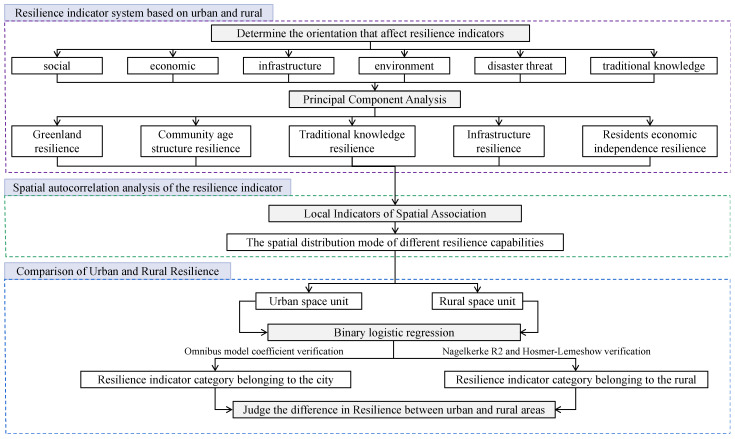
Research framework for comparing urban and rural resilience.

**Figure 2 ijerph-19-08911-f002:**
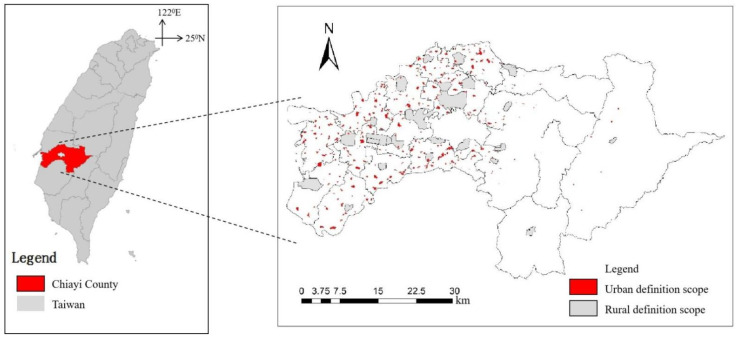
Research scope and spatial distribution of urban and rural areas.

**Figure 3 ijerph-19-08911-f003:**
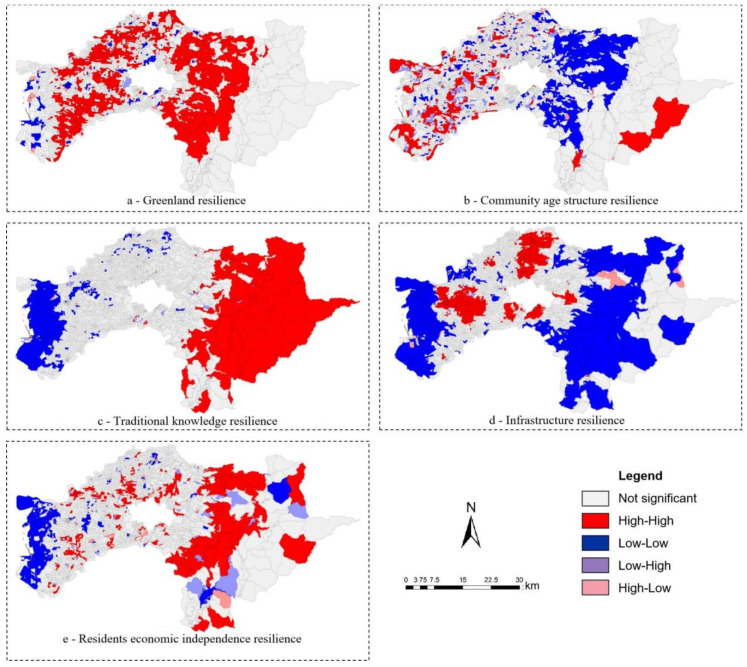
Spatial distribution mode of resilience abilities.

**Table 1 ijerph-19-08911-t001:** Indicator system and Principal Component Analysis.

Orientation	Indicator	Relation	Principal Component				
			1	2	3	4	5
Society	Aging index	−	0.156	0.482	0.046	0.235	−0.069
	Education level	+	−0.238	−0.739 *	0.085	0.297	0.084
	Household size	+	−0.093	−0.831 *	0.082	0.171	0.085
	Dependency ratio	−	0.054	0.702 *	−0.007	−0.031	0.033
	Population density	−	0.803 *	0.111	0.001	−0.105	−0.122
	Disabled population	−	0.041	0.499	−0.002	0.129	0.183
Economy	Agricultural land area	+	0.863 *	0.09	−0.157	0.012	0.124
	Number of agricultural households	+	−0.007	−0.152	−0.047	−0.131	0.19
	Residence income	+	−0.099	0.099	−0.036	0.616 *	−0.031
	Low-income households	−	−0.004	0.207	−0.106	0.09	0.508 *
Infrastructure	Medical facilities	+	−0.161	−0.018	−0.078	0.577 *	0.008
	School	+	−0.526 *	−0.172	−0.015	0.178	−0.114
	Fire station	+	−0.223	−0.099	−0.116	0.565 *	−0.155
	Road density	+	−0.722 *	0.018	−0.072	0.172	0.037
Environment	Impervious area	−	0.009	0.018	−0.291	−0.002	0.103
	Green infrastructure	+	0.157	−0.088	0.127	0.556 *	0.373
	Green area	+	−0.09	0.093	−0.469	0.232	−0.183
Disaster threat	Earth−rock flow potential	−	0.857 *	0.214	0.026	−0.07	−0.1
	Stratum subsidence	−	−0.057	0.075	−0.005	0.065	−0.730 *
	Landslides	−	0.886 *	0.047	0.075	0.009	0.184
Traditional knowledge	Percentage of indigenous population	+	−0.005	0.013	0.872 *	−0.015	−0.021
	Proportion of aboriginal elderly population	+	−0.019	0.033	0.841 *	0.018	−0.034
Eigenvalues			4.469	2.168	1.833	1.522	1.1
Measures of variation (%)			20.313	9.853	8.33	6.92	5.002
Cumulative explained variance ratio(%)			20.313	20.166	38.495	45.416	50.418
Kaiser−Meyer−Olkin (KMO)			0.767				
Bartlett’s sphericity test			Significance: 0.000; degree of freedom: 0.231				

Note: * indicates high correlation; + and − indicates the degree of positive and negative influence of data size.

**Table 2 ijerph-19-08911-t002:** Binary logistic regression model analysis.

Resilience Indicator		B	S.E.	Wald	Exp(B)	Significance	Possible Categories That Increase per Unit Volume
Pc4	Infrastructure resilience	1.339	0.082	264.721	3.817	0.000	Urban
Pc2	Community age structure resilience	0.694	0.105	43.680	2.003	0.000	Urban
Pc1	Greenland resilience	0.300	0.069	18.812	1.350	0.000	Urban
Pc5	Residents economic independence resilience	−0.398	0.088	20.386	0.671	0.000	Rural
Pc3	Traditional knowledge resilience	−0.422	0.149	7.994	0.655	0.005	Rural
Constant term		1.065	0.101	111.010	1	0.000	
Model sig = 0.000		Nagelkerke *R*^2^ = 0.411			Hosmer−lemeshow = 0.408 > 0.05		
Number of samples = 1645 (rural = 512, urban = 1133)							

Note: B is the estimated value of the regression coefficient; S.E. is the standard error; Wald is used to test the significance of the regression coefficient; df is the degree of freedom; Exp(B) is used to explain the meaning of the regression equation; Nagelkerke *R*^2^ represents the explanatory ability of the model; Hosmer–Lemeshow is mainly used to test the fit of the model.

## Data Availability

All data generated or analyzed during this study are included in this article.

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
