# Peer review of "A Comparative Study of the Resilience of Urban and Rural Areas under Climate Change"

_ijerph, 2022, doi:10.3390/ijerph19158911_

Round 1

Reviewer 1 Report

However, the title of the paper and topic are very interesting, the paper has a long way to go to become a scientific paper.

This paper has explored a very comprehensive topic from a very shallow perspective. While there are numerous issues, I mention only two of the most important ones. Many key terms have been mentioned and skipped in this paper. For example, “resiliency“ is defined in various ways and in each context has a different meaning. It is not clear what “Resilience” is. In addition, these types of studies need a comprehensive literature review. I can not see any literature review that is very strange for me. In addition, the research methodology that has been used is not acceptable. More justification needs to be discussed for the research methodology.

All in all, I can not recommend this paper, and I must reject it.

Author Response

Dear reviewer,

We have greatly appreciated the reviewers’ efforts. Comments and feedback were very constructive and able to improve the quality of the manuscript.

My Manuscript: ijerph-1791159

Article titled "A Comparative Study of the Resilience of Urban and Rural Areas under Climate Change"

Modified parts of the manuscript have been shown in red font. The following is a summary of the reviewers' comments.

Reviewer 1

Dear reviewer,

We have greatly appreciated the reviewers’ efforts. Comments and feedback were very constructive and able to improve the quality of the manuscript.

My Manuscript: ijerph-1791159

Article titled "A Comparative Study of the Resilience of Urban and Rural Areas under Climate Change"

Modified parts of the manuscript have been shown in red font. The following is a summary of the reviewers' comments.

  1. Many key terms have been mentioned and skipped in this paper. For example, “resiliency“ is defined in various ways and in each context has a different meaning. It is not clear what “Resilience” is.

Reply: Thanks to the reviewer for your suggestions, we have added the concept of resilience to this article, as shown in line 46-55.

“The field of ecology first introduced the concept of resilience, defining it as "a measure of the persistence of a system and its ability to absorb change and disturbance" [5]. Since then, many concepts of resilience have been derived. The concept of resilience is sometimes used as a measure of the ability of a system to recover after an event occurs, and sometimes it is regarded as the speed at which the system returns to its original state when an event occurs. However, it is generally the ability of internal systems to resist, buffer disturbances, absorb disturbances, self-organize, learn and adapt in response to external shock events [6, 7]. Therefore, this study focuses the concept of resilience on the ability of human systems to withstand the threat of external events under existing internal conditions.”

  1. these types of studies need a comprehensive literature review. I can not see any literature review that is very strange for me.

Reply: Thanks to the reviewer's suggestion, we have increased the content of the literature review accordingly to make it more complete. Specifically, such as line 70-95. We have also added a literature discussion section to the Discussion section, such as line 432-454.

“Cities have functions to meet the various needs of local residents, and climate change will cause inevitable impact on the functions that the urban system provides [10]. These functions include social and economic functions, infrastructure, environment, and anti-disaster facilities. This is also the main basis for the scholars to establish the indicators of resilience. However, for rural areas, in addition to their basic functions, the experience left over from history also plays an important role in coping with climate disasters [11]. Therefore, traditional knowledge is also a major factor in dealing with disasters.

In terms of resilience assessment, there is still a lack of consensus on the measurement methods and operations of resilience. The Disaster Resilience of Place Model proposed by Cutter et al. (2008) is mainly to quantify the spatial resilience of selected places, which can be classified as a local resilience model; The Coupled Social-Ecological Metrics model mainly introduces the tools of system dynamics and complexity to analyze the resilience of the community; The Teleconnection Metrics model mainly hopes to solve the problem of nonlinear dynamics in the Nested System, so it analyzes through different geographical locations to explore how the community is related to factors in different geographical locations through long-distance connections, but this method is mainly qualitative evaluation[12, 13]. Taken together, community resilience is deeply affected by vulnerability, global climate change, and natural disasters, resulting in each knowledge area having its own research framework and a set of ways to conceptualize community resilience. As a result, there is currently a lack of consensus on quantitative methods for assessing community resilience. Since this research hopes to establish resilience indicators that can be used to evaluate local urban and rural areas, the construction of the resilience indicators will be based on the local resilience indicators, and the advantages of various evaluation methods will be integrated to establish an evaluation model of urban and rural resilience.”

“In the previous research, Tobin (1999) took Florida as the research object, which mainly measured the degree of internal resilience of the state after the impact of Hurricane Andrew [31]. However, it does not have a corresponding practice site, but is based on qualitative analysis. This study combines local characteristics and provides the possibility to assess urban and rural resilience. Second, Mayunga's (2007) study further extended to various disaster types and established a Community Disaster Resilience Index [32]. However, in resilience assessment, it is mainly discussed from the perspective of economic capital, but it ignores resilience capabilities other than environmental or social capital. A major feature of this study is that it expands the concept of resilience and establishes an evaluation system for urban and rural resilience from social, economic, environmental and traditional knowledge. Then Yoon et al. (2016) also established six dimensions including people, society, economy, environment, disaster prevention system and urban space based on the Community Disaster Resilience Index, and conducted resilience measures for 229 basic self-governing groups in South Korea [6]. However, the blind spot lies in the failure to clearly identify the resilience of local communities for discussion, while this study is based on local resilience construction and has wide practical implementation. Research by Hudec, Reggiani, and Šiserová (2018) developed the Resilience Capacity Index to measure the resilience or rebound of regions from climate change shocks, but it also failed to clearly identify the resilience of local communities [8]. To sum up, there is a lack of application of urban and rural resilience indicators in previous resilience index assessments. The establishment of indicators for key factors affecting urban and rural resilience in this study is an important step to improve localized urban and rural resilience.”

Reviewer 2 Report

This is a relevant study that compares the differences in the resilience of urban and rural areas under climate change. The work is well written and is timely in as far as addressing the challenges posed by climate change and extreme weather events are concerned. Although, the definitions of urban and rural varies across different geographic contexts it is worth exploring how these areas are differently impacted and to also determine the key indicators for their resilience. However, I have made a few comments as follows:

Major comments:

·         The study needs to improve the cross referencing and citation as the issues under study are not new and there is a plethora of research studies in these issues. The discussion also needs to cite other studies with similar or contrary arguments.

Minor comments

·         Add ‘either’ in line 22 in sentence ending with ‘…concentrated in urban or rural.’

·         Ending of sentence in line 70 should read ‘…that spatial unit.’

·         What do the authors mean by smallest statistical areas in line 92? Is it the number of people? How about area size? This needs to be clarified.

·         Correct the legend in Figure 1, there is no black border around Taiwan. Please correct the legend.

·         Line 109 is not true, there are other statistical softwares that can be used to conduct PCA analysis such as XLSTAT, Minitab, STATISTICA etc.

·         Also, write SPSS in full when used for the first time.

·          What does line 179 mean, what is the significance standard?

·         Line 136 to 138 is not clear and confusing. You should just describe the type of outcome variable and why a logistic regression is suitable. Alluding to a linear regression here does not make sense since it is not even explained as to under what circumstances should it be used.

·         The Research Framework in 2.3 should be 2.1 as it is the one that details the approach followed by the study.

·         Write KMO in full first in line 224.

·         In line 294, it is confusing that the r square is at 41% and the significance level is at 40%. In my opinion, this is a poor fit model. Can the authors explain how is it that the 40% is less than the 5% significance level in line 295?

Author Response

Reviewer 2

Dear reviewer,

We have greatly appreciated the reviewers’ efforts. Comments and feedback were very constructive and able to improve the quality of the manuscript.

My Manuscript: ijerph-1791159

Article titled "A Comparative Study of the Resilience of Urban and Rural Areas under Climate Change"

Modified parts of the manuscript have been shown in red font. The following is a summary of the reviewers' comments.

  1. This is a relevant study that compares the differences in the resilience of urban and rural areas under climate change. The work is well written and is timely in as far as addressing the challenges posed by climate change and extreme weather events are concerned. Although, the definitions of urban and rural varies across different geographic contexts it is worth exploring how these areas are differently impacted and to also determine the key indicators for their resilience.

Reply: Thanks to the reviewer for your affirmation, your suggestions are of great help to the improvement of the article.

  1. The study needs to improve the cross referencing and citation as the issues under study are not new and there is a plethora of research studies in these issues. The discussion also needs to cite other studies with similar or contrary arguments.

Reply: Thanks to the reviewer for your suggestions. We have added a comprehensive literature review to the introduction to make the research question more complete, as specified in lines 46-55, 78-95. At the same time, in the discussion section, we have added the corresponding opinions of different authors and the differences of this article, such as line 432-477.

line 46-55 “The field of ecology first introduced the concept of resilience, defining it as "a measure of the persistence of a system and its ability to absorb change and disturbance" [5]. Since then, many concepts of resilience have been derived. The concept of resilience is sometimes used as a measure of the ability of a system to recover after an event occurs, and sometimes it is regarded as the speed at which the system returns to its original state when an event occurs. However, it is generally the ability of internal systems to resist, buffer disturbances, absorb disturbances, self-organize, learn and adapt in response to external shock events [6, 7]. Therefore, this study focuses the concept of resilience on the ability of human systems to withstand the threat of external events under existing internal conditions.”

line78-95 “In terms of resilience assessment, there is still a lack of consensus on the measurement methods and operations of resilience. The Disaster Resilience of Place Model proposed by Cutter et al. (2008) is mainly to quantify the spatial resilience of selected places, which can be classified as a local resilience model; The Coupled Social-Ecological Metrics model mainly introduces the tools of system dynamics and complexity to analyze the resilience of the community; The Teleconnection Metrics model mainly hopes to solve the problem of nonlinear dynamics in the Nested System, so it analyzes through different geographical locations to explore how the community is related to factors in different geographical locations through long-distance connections, but this method is mainly qualitative evaluation[12, 13]. Taken together, community resilience is deeply affected by vulnerability, global climate change, and natural disasters, resulting in each knowledge area having its own research framework and a set of ways to conceptualize community resilience. As a result, there is currently a lack of consensus on quantitative methods for assessing community resilience. Since this research hopes to establish resilience indicators that can be used to evaluate local urban and rural areas, the construction of the resilience indicators will be based on the local resilience indicators, and the advantages of various evaluation methods will be integrated to establish an evaluation model of urban and rural resilience.”

line 432-477 “In the previous research, Tobin (1999) took Florida as the research object, which mainly measured the degree of internal resilience of the state after the impact of Hurricane Andrew [31]. However, it does not have a corresponding practice site, but is based on qualitative analysis. This study combines local characteristics and provides the possibility to assess urban and rural resilience. Second, Mayunga's (2007) study further extended to various disaster types and established a Community Disaster Resilience Index [32]. However, in resilience assessment, it is mainly discussed from the perspective of economic capital, but it ignores resilience capabilities other than environmental or social capital. A major feature of this study is that it expands the concept of resilience and establishes an evaluation system for urban and rural resilience from social, economic, environmental and traditional knowledge. Then Yoon et al. (2016) also established six dimensions including people, society, economy, environment, disaster prevention system and urban space based on the Community Disaster Resilience Index, and conducted resilience measures for 229 basic self-governing groups in South Korea [6]. However, the blind spot lies in the failure to clearly identify the resilience of local communities for discussion, while this study is based on local resilience construction and has wide practical implementation. Research by Hudec, Reggiani, and Šiserová (2018) developed the Resilience Capacity Index to measure the resilience or rebound of regions from climate change shocks, but it also failed to clearly identify the resilience of local communities [8]. To sum up, there is a lack of application of urban and rural resilience indicators in previous resilience index assessments. The establishment of indicators for key factors affecting urban and rural resilience in this study is an important step to improve localized urban and rural resilience.

Regarding the formulation of strategies for cities to respond to climate change, most of the previous research focused on reducing the impact of climate change through infrastructure planning [33, 34]. The analysis results of this study also confirmed the importance of infrastructure to urban resilience, but on the other hand, it was also found that green space is also one of the key factors of urban resilience. In fact, green space in the city used to be regarded as the land without economic value, but it has become a key factor affecting urban resilience under the threat of climate change. Therefore, how to increase the proportion of urban green space can be used as one of the important strategies for each city and county to establish spatial planning. Although it is also emphasized that infrastructure is an essential strategy for improving rural resilience, the long-term lack of a comprehensive planning strategy in rural area makes it difficult to increase the construction of infrastructure. According to the analysis results of this study, the resilience abilities of the rural areas mainly depend on the economic independence of local residents and traditional knowledge at non-hardware level. Therefore, in the spatial planning, how to effectively use these abilities is the key to the subsequent overall planning of rural resilience, and it is also the foundation of maintaining rural resilience before the infrastructure is improved in rural areas. To sum up, when planning the location of infrastructure in the future, it will help to reduce the damage caused by the city if we can consider the characteristics of infrastructure for the city in coping with climate change and pay attention to its co-benefits. Villages have long lacked sound planning strategies, and should rely more on their own resilience (such as traditional knowledge) to cope with climate change.”

  1. Add ‘either’ in line 22 in sentence ending with ‘…concentrated in urban or rural.’

Reply: Thanks to the reviewer for your patient discovery. We have added to the article.

  1. Ending of sentence in line 70 should read ‘…that spatial unit.’

Reply: Thanks to the reviewer for your patient discovery. We have added to the article.

  1. What do the authors mean by smallest statistical areas in line 92? Is it the number of people? How about area size? This needs to be clarified.

Reply: Thanks to the reviewer for your suggestions. We have added corresponding concepts, as shown in lines 154-160.

“In terms of the selection of cities, the urban planning district of Chiayi County was used as the standard, which defines the scope of the metropolitan area. Therefore, we select its community boundary as the statistical unit. Regarding the selection of villages, the rural area in the non-urban land use zoning was used as the standard. Its original definition standard is based on the current situation of use. Therefore, the minimum statistical area is mainly based on the community boundaries and village boundaries defined by the government. ”

  1. Correct the legend in Figure 1, there is no black border around Taiwan. Please correct the legend.

Reply: Thanks to the reviewer for your patient discovery. We have made corresponding changes.

  1. Line 109 is not true, there are other statistical softwares that can be used to conduct PCA analysis such as XLSTAT, Minitab, STATISTICA etc.

Reply: Thanks to the reviewer for your suggestions. We modified the corresponding statement. Such as line 177-178.

“ We used Statistical Product and Service Solutions (SPSS) software for analysis.”

  1. Also, write SPSS in full when used for the first time.

Reply: Thanks to the reviewer for your suggestions. Statistical Product and Service Solutions (SPSS) has been added to the text.

  1. Line 136 to 138 is not clear and confusing. You should just describe the type of outcome variable and why a logistic regression is suitable. Alluding to a linear regression here does not make sense since it is not even explained as to under what circumstances should it be used.

Reply: We removed this sentence, as you said it didn't help the article. “The greatest difference between the logistic regression model and the linear regression model depends on whether the analysis variable (dependent variable) is binary or dichotomous.”

  1. The Research Framework in 2.3 should be 2.1 as it is the one that details the approach followed by the study.

Reply: Thanks to the reviewers for their suggestions. We made an order adjustment.

  1. Write KMO in full first in line 224.

Reply:Thanks to the reviewer for the reminder. We have revised relevant statements such as "Kaiser-Meyer-Olkin (KMO)"

  1. In line 294, it is confusing that the r square is at 41% and the significance level is at 40%. In my opinion, this is a poor fit model. Can the authors explain how is it that the 40% is less than the 5% significance level in line 295?

Reply: Thanks to the reviewer for the reminder. We have added corresponding explanations to the article, such as line 346-351.

 “The accuracy of the binary logistic regression model used in this study is 41.1% (Nagelkerke R2=0.411, indicating that the predictive analysis accuracy of this model was the best 41.1%. The Hosmer-Lemeshow test is mainly used to test the fit of the model. The significance of the test result in this study is 0.408>0.05, which means that the hypothesis of good fit of the regression model is accepted. ”

Reviewer 3 Report

Chang et al. collected indicators of resilience through literature review, and used PCA, LISA, and Binary logistic regression to analyze the resilience of urban and rural under climate change. The research is interesting, but there still exist some questions unclear, which are listed as the followings.

1. It seems that the variables you used are collected at different years and you didn’t consider the time dimension in your experiments, but some variables like aging index, population, income, etc. would change year by year. Will the results change if you used the variables collected in different years? I think it would be better if you can add discussion of the uncertainties of your results caused by the datasets you use.

2. How do you distinguish urban and ruralWhat about rural-urban continuum?

3. Although you list variables information in supplement table 1, you didn’t cite it in the manuscript's main text. And it would be better to clarify what data you used in the methodology section.

4. Supplement Table 1: More information of datasets should be added. For example, what format is the data (shapefile?), the time window of the data (when the data is collected, or start year and end year).

5. All your results are based on correlation, but correlation is not causality. It seems that “causal inference” methods would be more helpful for the analysis rather than the correlation-based methods. At least, clarify the limitation of correlation in the manuscript.

6. L102-106: PCA isn’t a method proposed by this paper, therefore, citations should be added here.

7. L122: Rewrite the sentences here. The meanings of the variables are not explained clearly. what is Xi? What does “the value of z” mean?

8. L150: Should Bi be corrected to bi?

9. Figure3 subplot should be noted as (a)-(e)

10. Section 3.2: The results in Figure 3 should be described with more details.

11.L290: ACRGIS? Or ArcGIS?

Author Response

Reviewer 3

Dear reviewer,

We have greatly appreciated the reviewers’ efforts. Comments and feedback were very constructive and able to improve the quality of the manuscript.

My Manuscript: ijerph-1791159

Article titled "A Comparative Study of the Resilience of Urban and Rural Areas under Climate Change"

Modified parts of the manuscript have been shown in red font. The following is a summary of the reviewers' comments.

  1. Chang et al. collected indicators of resilience through literature review, and used PCA, LISA, and Binary logistic regression to analyze the resilience of urban and rural under climate change. The research is interesting, but there still exist some questions unclear, which are listed as the followings.

Reply: Thanks for the affirmation of the reviewer, your suggestions are of great help to the improvement of the article.

  1. It seems that the variables you used are collected at different years and you didn’t consider the time dimension in your experiments, but some variables like aging index, population, income, etc. would change year by year. Will the results change if you used the variables collected in different years? I think it would be better if you can add discussion of the uncertainties of your results caused by the datasets you use.

Reply: Thanks to the reviewer for the reminder. We add this to our research shortfalls. Such as line 486-491.

“Since the data were published in different years, the analysis based on the latest related data cannot facilitate the comparison of the changes in urban and rural resilience in terms of time under the influence of climate change. If more time-section data can be used, it will help to obtain more accurate resilience capabilities, and thus deepen the understanding of the differences in resilience between urban and rural areas.”

  1. How do you distinguish urban and rural?What about rural-urban continuum?

Reply: Thanks to the reviewer for your suggestions. We have added a description of the corresponding urban-rural classification, as shown in lines 154-160. At the same time, the classification of urban and rural areas is very complicated, and we can only divide it into a system of dual structure of urban and rural areas, so some research deficiencies have been added to the text to avoid readers misusing the broad applicability of the conclusions, such as line 491-494.

line 154-160 “In terms of the selection of cities, the urban planning district of Chiayi County was used as the standard, which defines the scope of the metropolitan area. Therefore, we select its community boundary as the statistical unit. Regarding the selection of villages, the rural area in the non-urban land use zoning was used as the standard. Its original definition standard is based on the current situation of use. Therefore, the minimum statistical area is mainly based on the community boundaries and village boundaries defined by the government. ”

line 491-494 “The spatial units compared in this study are limited to urban and rural areas formulated in the research design, and many areas that are not located in the above spatial units are ignored. Therefore, it is impossible to compare various types of national land spaces.”

  1. Although you list variables information in supplement table 1, you didn’t cite it in the manuscript's main text.

Reply: Thanks to the reviewer for the reminder. We have added corresponding marks in the text, such as line 260. “(see Supplemental table 1 for specific calculation method and data sources of indicators).”

  1. Supplement Table 1: More information of datasets should be added. For example, what format is the data (shapefile?), the time window of the data (when the data is collected)

Reply: Thanks to the reviewer for the reminder. We added the corresponding data types and statistics time. For details, see "Supplemental table 1"

  1. All your results are based on correlation, but correlation is not causality. It seems that “causal inference” methods would be more helpful for the analysis rather than the correlation-based methods. At least, clarify the limitation of correlation in the manuscript.

Reply: Thanks to the reviewer for the reminder. We add this to the research gap. Specifically as shown in line 494-497.

“(III) The results of this study are obtained based on correlations and have certain limitations. Follow-up research can also help analyze the presentation of results from the "causal inference" approach.”

  1. L102-106: PCA isn’t a method proposed by this paper, therefore, citations should be added here.

Reply: Thanks to the reviewer for the reminder. We have added the corresponding references.

  1. L122: Rewrite the sentences here. The meanings of the variables are not explained clearly. what is Xi? What does “the value of z” mean?

Reply: Thanks to the reviewer for the reminder. We have added corresponding explanations. Such as line 190-191.

“z represents the spatial attribute relationship between two adjacent regions;”

  1. L150: Should Bi be corrected to bi?

Reply: Thanks to the reviewer for your patient discovery. We have made corresponding changes.

  1. Figure3 subplot should be noted as (a)-(e).

Reply: Thanks to the reviewer for your patient discovery. We have made corresponding changes.

  1. Section 3.2: The results in Figure 3 should be described with more details.

Reply: Thanks to the reviewer for your suggestions. We have revised the title of Figure 3 to make it more concise. And we added a detailed description of Figure 3, as lines 319-340.

“In terms of the spatial distribution pattern of greenland resilience (Figure 3-a), most of the non-mountainous and coastal areas have high resilience scores of spatial units and adjacent spatial units. In addition, among the space units in the coastal area, many space units and their adjacent space units have low values. In the resilience score of community age structure resilience (Figure 3-b), most of the regions with high values of spatial units and adjacent spaces are located in the west, and some spatial units in the eastern mountainous areas also exhibit such characteristics. In the resilience score of traditional knowledge capacity (Fig. 3-c), the spatial units in the eastern mountainous area and the adjacent spatial units show a distribution pattern of high value concentration. In addition, the spatial distribution patterns with low toughness scores and adjacent spatial units with low values are mostly located in the western coastal areas. In the score of infrastructure resilience (Fig. 3-d), the areas with high values of both spatial units and adjacent spatial units are mostly concentrated in the urban areas near the west. The spatial units with lower resilience scores and the adjacent spatial units with low values are located in the western coastal and eastern hilly areas. Among the resilience scores of residents economic independence ability (Figure 3-e), those with high resilience scores and their adjacent spatial units are mostly concentrated in the eastern non-mountainous areas. Those with low resilience scores and their adjacent spatial units are mostly located in the western coastal areas. The analysis results of spatial correlation showed that there may be differences in the resilience abilities between cities and villages. Therefore, it is necessary to further explore the differences between urban and rural resilience.”

11.L290: ACRGIS? Or ArcGIS?

Reply: Thanks to the reviewer for your patient discovery. We have made corresponding changes.

Reviewer 4 Report

1- Abstract require some quantified results

2- The introduction should be improved to highlight the hypothesis and novelty using additional references

3- In figure 3, expand the caption by explaining the panels in brief

4- Discussion section need to be improved 

5- Conclusion is too long, please minimize it and remove references from Conclusion to be in Discussion 

Author Response

Reviewer 4

Dear reviewer,

We have greatly appreciated the reviewers’ efforts. Comments and feedback were very constructive and able to improve the quality of the manuscript.

My Manuscript: ijerph-1791159

Article titled "A Comparative Study of the Resilience of Urban and Rural Areas under Climate Change"

Modified parts of the manuscript have been shown in red font. The following is a summary of the reviewers' comments.

  1. Abstract require some quantified results.

Reply: Thanks to the reviewer for your suggestions. We have added eg lines 22-26.

“Binary logistic regression was performed, and the results showed urban areas have more prominent abilities in infrastructure resilience (The coefficient value is 1.339), community age structure resilience (0.694), and greenland resilience (0.3), while rural areas are more prominent in terms of the residents economic independence resilience (-0.398) and traditional knowledge resilience (-0.422).”

  1. The introduction should be improved to highlight the hypothesis and novelty using additional references.

Reply: Thanks to the reviewer for your suggestions. We have added corresponding content in the introduction, such as adding an explanation of the concept of "resilience" and the definition of my literature, and adding a literature review to expand the novelty and difference of the article. Such as line 46-55, 78-95.

line 46-55 “The field of ecology first introduced the concept of resilience, defining it as "a measure of the persistence of a system and its ability to absorb change and disturbance" [5]. Since then, many concepts of resilience have been derived. The concept of resilience is sometimes used as a measure of the ability of a system to recover after an event occurs, and sometimes it is regarded as the speed at which the system returns to its original state when an event occurs. However, it is generally the ability of internal systems to resist, buffer disturbances, absorb disturbances, self-organize, learn and adapt in response to external shock events [6, 7]. Therefore, this study focuses the concept of resilience on the ability of human systems to withstand the threat of external events under existing internal conditions.”

line 78-95 “In terms of resilience assessment, there is still a lack of consensus on the measurement methods and operations of resilience. The Disaster Resilience of Place Model proposed by Cutter et al. (2008) is mainly to quantify the spatial resilience of selected places, which can be classified as a local resilience model; The Coupled Social-Ecological Metrics model mainly introduces the tools of system dynamics and complexity to analyze the resilience of the community; The Teleconnection Metrics model mainly hopes to solve the problem of nonlinear dynamics in the Nested System, so it analyzes through different geographical locations to explore how the community is related to factors in different geographical locations through long-distance connections, but this method is mainly qualitative evaluation[12, 13]. Taken together, community resilience is deeply affected by vulnerability, global climate change, and natural disasters, resulting in each knowledge area having its own research framework and a set of ways to conceptualize community resilience. As a result, there is currently a lack of consensus on quantitative methods for assessing community resilience. Since this research hopes to establish resilience indicators that can be used to evaluate local urban and rural areas, the construction of the resilience indicators will be based on the local resilience indicators, and the advantages of various evaluation methods will be integrated to establish an evaluation model of urban and rural resilience.”

  1. In figure 3, expand the caption by explaining the panels in brief.

Reply: Thanks to the reviewer for your suggestions. We have added a detailed description and simplified title of Figure 3. Such as line 319-340.

“In terms of the spatial distribution pattern of greenland resilience (Figure 3-a), most of the non-mountainous and coastal areas have high resilience scores of spatial units and adjacent spatial units. In addition, among the space units in the coastal area, many space units and their adjacent space units have low values. In the resilience score of community age structure resilience (Figure 3-b), most of the regions with high values of spatial units and adjacent spaces are located in the west, and some spatial units in the eastern mountainous areas also exhibit such characteristics. In the resilience score of traditional knowledge capacity (Fig. 3-c), the spatial units in the eastern mountainous area and the adjacent spatial units show a distribution pattern of high value concentration. In addition, the spatial distribution patterns with low toughness scores and adjacent spatial units with low values are mostly located in the western coastal areas. In the score of infrastructure resilience (Fig. 3-d), the areas with high values of both spatial units and adjacent spatial units are mostly concentrated in the urban areas near the west. The spatial units with lower resilience scores and the adjacent spatial units with low values are located in the western coastal and eastern hilly areas. Among the resilience scores of residents economic independence ability (Figure 3-e), those with high resilience scores and their adjacent spatial units are mostly concentrated in the eastern non-mountainous areas. Those with low resilience scores and their adjacent spatial units are mostly located in the western coastal areas. The analysis results of spatial correlation showed that there may be differences in the resilience abilities between cities and villages. Therefore, it is necessary to further explore the differences between urban and rural resilience.”

  1. Discussion section need to be improved.

Reply: In the Discussion section, we have added corresponding views of different authors, differences in this article, and inadequacies of the article. For example, line 432-454, 471-476, 486-497. It mainly highlights the innovation and adaptability of the article by comparing with previous studies.

line 432-454 “In the previous research, Tobin (1999) took Florida as the research object, which mainly measured the degree of internal resilience of the state after the impact of Hurricane Andrew [31]. However, it does not have a corresponding practice site, but is based on qualitative analysis. This study combines local characteristics and provides the possibility to assess urban and rural resilience. Second, Mayunga's (2007) study further extended to various disaster types and established a Community Disaster Resilience Index [32]. However, in resilience assessment, it is mainly discussed from the perspective of economic capital, but it ignores resilience capabilities other than environmental or social capital. A major feature of this study is that it expands the concept of resilience and establishes an evaluation system for urban and rural resilience from social, economic, environmental and traditional knowledge. Then Yoon et al. (2016) also established six dimensions including people, society, economy, environment, disaster prevention system and urban space based on the Community Disaster Resilience Index, and conducted resilience measures for 229 basic self-governing groups in South Korea [6]. However, the blind spot lies in the failure to clearly identify the resilience of local communities for discussion, while this study is based on local resilience construction and has wide practical implementation. Research by Hudec, Reggiani, and Šiserová (2018) developed the Resilience Capacity Index to measure the resilience or rebound of regions from climate change shocks, but it also failed to clearly identify the resilience of local communities [8]. To sum up, there is a lack of application of urban and rural resilience indicators in previous resilience index assessments. The establishment of indicators for key factors affecting urban and rural resilience in this study is an important step to improve localized urban and rural resilience.”

line 471-476 “To sum up, when planning the location of infrastructure in the future, it will help to reduce the damage caused by the city if we can consider the characteristics of infrastructure for the city in coping with climate change and pay attention to its co-benefits. Villages have long lacked sound planning strategies, and should rely more on their own resilience (such as traditional knowledge) to cope with climate change.”

line 486-497 “There are still some limitations in this research. (I) Since the data were published in different years, the analysis based on the latest related data cannot facilitate the comparison of the changes in urban and rural resilience in terms of time under the influence of climate change. If more time-section data can be used, it will help to obtain more accurate resilience capabilities, and thus deepen the understanding of the differences in resilience between urban and rural areas. (Ⅱ) The spatial units compared in this study are limited to urban and rural areas formulated in the research design, and many areas that are not located in the above spatial units are ignored. Therefore, it is impossible to compare various types of national land spaces. (III) The results of this study are obtained based on correlations and have certain limitations. Follow-up research can also help analyze the presentation of results from the "causal inference" approach.”

  1. Conclusion is too long, please minimize it and remove references from Conclusion to be in Discussion.

Reply: Thanks to the reviewers for their patient discovery. We have made corresponding changes. Changed from 473 words to 281 words.

Round 2

Reviewer 1 Report

This paper needs a major revision.

The current literature review is not acceptable at all. A new section should be provided, and relevant studies should be critically reviewed. This research area is very solid.

it is better to use "extreme weather events" instead of "extreme weather disasters."

Although climate change has a wide range of impacts on humans and nature,.... it is not clear what the aim of the authors is. in the introduction, they should make it clear what aspect of climate change is analysed in this study.

How do the authors reach the indicators? line 112 is not acceptable for this research at all! See: The Integration of Lean and Resilience Paradigms: A Systematic Review Identifying Current and Future Research Directions.

Each of the indicators should be explained, and several examples should be provided for them. For example, for disaster treats a wide range of disaster risk response strategies are available in the literature, see: An integrated decision model for managing hospital evacuation in response to an extreme flood event: A case study of the Hawkesbury‐Nepean River, NSW, Australia and A modelling framework to design an evacuation support system for healthcare infrastructures in response to major flood events and Hospital evacuation modelling: A critical literature review on current knowledge and research gaps

The quality of figure 1 is very low. it should be improved.

a better heading can be selected for "2.3."

please revise line 183.

Author Response

Reviewer 1

Dear reviewer,

We have greatly appreciated the reviewers’ efforts. Comments and feedback were very constructive and able to improve the quality of the manuscript.

My Manuscript: ijerph-1791159

Article titled "A Comparative Study of the Resilience of Urban and Rural Areas under Climate Change"

Modified parts of the manuscript have been shown in red font. The following is a summary of the reviewers' comments.

Reply to reviewer:

  1. it is better to use "extreme weather events" instead of "extreme weather disasters."

Reply:Thanks to the reviewer's suggestion, we have replaced "extreme weather disasters." with "extreme weather events."

  1. Although climate change has a wide range of impacts on humans and nature,.... it is not clear what the aim of the authors is. in the introduction, they should make it clear what aspect of climate change is analysed in this study.

Reply: Thanks to the reviewer's suggestion, we have increased the research content of this study on climate change accordingly. Such as line 54-69.

“However, the previous discussion on the resilience strategy to cope with climate change mainly focuses on urban spaces or densely populated areas [8], while ignoring rural areas that are also threatened by climate change. Therefore, there is still a lack of understanding of the risks and corresponding strategy in rural areas. In fact, cities and rural areas have significantly different resilience in society, economy, environment and infrastructure when facing disaster risks caused by climate change [9]. However, most of the previous studies on urban resilience explored the ability of the economy and industry of urban system to respond to the threat posed by climate change, while most of the studies on rural resilience focused on the ability of the rural areas to survive under climate change. The external events will bring the same impact to both the urban and rural areas. Therefore, building an indicator framework of resilience that adapts to the cities and villages will help to identify the difference in resilience between the urban area and rural area, and will facilitate the future allocation of related resources and improve strategies. The research on urban and rural resilience is also a strategic exploration to cope with the different responses of urban and rural areas to disasters and corresponding mechanisms under climate change.”

  1. How do the authors reach the indicators? line 112 is not acceptable for this research at all! See: The Integration of Lean and Resilience Paradigms: A Systematic Review Identifying Current and Future Research Directions.

Reply: Thanks to the reviewer for such an excellent article. We followed the way the indicators were written and included them in the references for this study. Such as line 220-228.

“With reference to the existing literature and the framework of CIMO (context-intervention-mechanism-outcome), and based on the principles of scientificity, objectivity, comprehensiveness and data availability, this study initially established an urban and rural disaster resilience indicator system[23]. In order to eliminate the subjectivity of index selection decision, we refer to CNKI, Web of Science and ScienceDirect three databases, and select index elements with high frequency in recent years. These indicators are further selected by consultants and government staff in the fields of resilient cities, disaster risk, and urban-rural development, and are constructed in two dimensions: urban resilience and rural resilience (specific indicators are shown in Supplemental table 1).”

  1. Each of the indicators should be explained, and several examples should be provided for them. For example, for disaster treats a wide range of disaster risk response strategies are available in the literature, see: An integrated decision model for managing hospital evacuation in response to an extreme flood event: A case study of the Hawkesbury‐Nepean River, NSW, Australia and A modelling framework to design an evacuation support system for healthcare infrastructures in response to major flood events and Hospital evacuation modelling: A critical literature review on current knowledge and research gaps

Reply: Thanks to the reviewers for such an excellent article. We have added relevant references in "3.1. The construction of the indicator system and Principal Component Analysis", and added some case studies to support the establishment of this indicator. Such as line 239-241, 244-246, 259-262.

line 239-241 “ For example, Yoon et al. (2016) established six dimensions, including people, society, economy, environment, disaster prevention system, and urban space, and then assessed the resilience of communities in the face of disasters [6].

line 244-246 “For example, Maziar Yazdani et al. studied the impact of flood risk on medical infrastructure and proposed a new modeling framework to improve the resilience of medical infrastructure to floods [28, 29].”

line 259-262 “ For example, The study of Altieri et al. (2017) found that traditional knowledge can effectively reduce the impact of climate change for local residents and maintain a certain amount of agricultural products. This also means that traditional knowledge for villages will help improve their resilience [11]. ”

  1. The quality of figure 1 is very low. it should be improved.

Reply: We replaced the higher-resolution figure.

  1. a better heading can be selected for "2.3."

Reply: We changed "method" to "A comparative approach to urban-rural resilience differences"

  1. please revise line 183.

Reply: We have revised relevant statements such as line 184-186.

“This study uses LISA to identify the spatial distribution patterns of different resilience abilities, and then explores the spatial differences of different resilience abilities. ”

Reviewer 3 Report

Thanks for your consideration of my review. In general, your response is suitable for my comments. I suggest accepting the manuscript.

Author Response

reviewer 3.
1. Thanks for your consideration of my review. In general, your response is suitable for my comments. I suggest accepting the manuscript.
reply: Thank you for your affirmation of my revision, I am honored for your guidance on the article.